# Alternative Options for Skin Cancer Therapy via Regulation of AKT and Related Signaling Pathways

**DOI:** 10.3390/ijms21186869

**Published:** 2020-09-18

**Authors:** Sun-Young Hwang, Jung-Il Chae, Ah-Won Kwak, Mee-Hyun Lee, Jung-Hyun Shim

**Affiliations:** 1College of Korean Medicine, Dongshin University, Naju, Jeonnam 58245, Korea; tigger5368@naver.com; 2Department of Dental Pharmacology, School of Dentistry and Institute of Oral Bioscience, Jeonbuk National University, Jeonju 54896, Korea; jichae@jbnu.ac.kr; 3Department of Pharmacy, College of Pharmacy, Mokpo National University, Jeonnam 58554, Korea; rhkrdkdnjs12@mokpo.ac.kr; 4Department of Biomedicine, Health & Life Convergence Sciences, BK21 FOUR, Mokpo National University, Jeonnam 58554, Korea

**Keywords:** AKT, skin cancer, AKT inhibitor, signal transduction

## Abstract

Global environmental pollution has led to human exposure to ultraviolet (UV) radiation due to the damaged ozone layer, thereby increasing the incidence and death rate of skin cancer including both melanoma and non-melanoma. Overexpression and activation of V-akt murine thymoma viral oncogene homolog (AKT, also known as protein kinase B) and related signaling pathways are major factors contributing to many cancers including lung cancer, esophageal squamous cell carcinoma and skin cancer. Although BRAF inhibitors are used to treat melanoma, further options are needed due to treatment resistance and poor efficacy. Depletion of AKT expression and activation, and related signaling cascades by its inhibitors, decreases the growth of skin cancer and metastasis. Here we have focused the effects of AKT and related signaling (PI3K/AKT/mTOR) pathways by regulators derived from plants and suggest the need for efficient treatment in skin cancer therapy.

## 1. Introduction

Skin cancer is one of the most frequent cancers worldwide [1,2], with increasing annual costs and morbidity [3,4]. Based on their cellular origin, skin cancers are mainly divided into melanoma (derived from melanocytes) and non-melanoma skin cancer (NMSC, from epithelial cells). NMSC includes basal cell carcinoma (~80%) and squamous cell carcinoma (~16%) and melanoma accounts for ~4% of all skin cancer [5]. Even though it accounts for a small portion of skin cancer, melanoma is the most malignant cause of invasive metastasis with a five year survival rate of ~20% [6]. An estimated 7,700,000 new cases of NMSC, including 5,900,000 basal cell carcinomas and 1,800,000 squamous cell carcinomas, have been reported globally including 195 countries accounting for 65,000 cancer deaths [2,7]. An estimated ~100,350 new cases of melanoma and 6850 related deaths have been reported in the USA during 2020 [1].

The major environmental risk factors for skin cancer (melanoma and NMSC) include UVA, UVB and UVC [6,8]. UVC has the shortest wavelengths spectrum (100–280 nm) and after absorption by the ozone layer and atmosphere, it barely reaches the earth. Generally, the human skin is affected by solar UV (100–400 nm) radiation composed of UVA (315–400 nm, ~95%) and UVB (280–315 nm, ~5%) rays triggering skin inflammation, carcinogenesis and cancer development [9,10].

Genetic mutation of p53, BRAF, RAS, CDKN2A and PTEN, and abnormal expression/activation of T-LAK cell-originated protein kinase (TOPK), mitogen-activated protein kinase kinase (MEK), 90 kDa ribosomal S6 kinase (RSK) and AKT in melanoma and NMSC induce cancer cell signal transduction thereby promoting skin carcinogenesis and cancer cell proliferation, migration and invasion [6,9,11,12,13,14]. Solar UV triggers PTEN mutations. Other environmental stimuli induce overexpression and/or activation of AKT and related signaling pathways in skin cancer proliferation and survival. Therefore, it is not surprising that AKT-mediated signaling is one of the main mechanisms in skin cancer therapy. The focus of the article is on AKT and related signaling pathways, and their implications in skin cancer therapy.

Strategies for the skin cancer management include surgery, radiation, chemotherapy and cutting-edge targeted therapies [13,15]. Dacarbazine, temozolomide, paclitaxel, cisplatin and carboplatin are generally used to treat melanoma, whereas vismodegib, sonidegib and cetuximab are used to target basal and squamous cell carcinoma. Vemurafenib is a competitive inhibitor of BRAF kinase activity, and especially inhibits melanoma with a V600 mutation. However, treatment with these therapeutic agents is associated with severe side effects and a low quality of life including nausea, hair loss and increased risk of infection. It also increases the risk of resistance immediately after treatment for several months. Therefore, natural compounds are the preferred choice due to their reduced risk of toxicity and cost-effectiveness.

## 2. AKT and Related Signaling Pathways Are Important in Skin Cancer Regulation

AKT is a serine/threonine-specific protein kinase and alternatively named in protein kinase B (PKB). It plays a key role in multiple cellular processes such as protein synthesis, cell proliferation, cell cycle progression, survival and migration as well as metabolism (Figure 1) [16,17]. The AKT subtypes include AKT1, AKT2 and AKT3 which contain conserved domains such as pleckstrin homology (PH) domain for protein–protein interaction or interaction with phosphatidylinositol phosphate (PIP)-containing lipid bilayers, linker, kinase domain for catalytic activity and regulatory (hydrophobic motif, HM) domain for compete activation with high homology (~80%) as shown in Figure 2. Most of the direct AKT inhibitors target the ATP binding pocket in the kinase and HM domain. In the AKT knock-out (KO) mouse model, each AKT isoform showed different phenotypes such as growth retardation and altered placental development phenotype with increasing lethality in AKT1 KO, insulin resistance and mild growth retardation in AKT2 KO mice, and abnormal development of skin and bone/muscle defect in AKT3 KO [18,19]. AKT is highly activated and expressed in cancers including lung, ovarian, pancreatic and esophageal squamous cell carcinoma [16,20]. In skin cancer, the active form of phospho-AKT (pAKT) was overexpressed in 22 (54%) of 41 benign nevi and 112 (71.3%) of 157 primary melanoma tumors compared to normal adjacent tissue, resulting in a reciprocal five-year survival rate in human metastatic melanoma [21,22]. Together with downstream target protein mTOR, the activated AKT1 induces highly metastatic melanomas involving lung (67%) and brain (17%) in BRAF^V600E^/Cdkn2a^Null^ mice [23]. Thus, AKT is implicated in melanoma metastasis. AKT protein is activated by the phosphorylation of PI3K (p110/p85 heterodimer), which recruits AKT to the cell membrane through phosphatidylinositol (3,4,5)-trisphosphate (PIP3)-PH domain binding. It’s also stimulated by receptor tyrosine kinase (RTK), phosphatidylinositol-dependent kinase (PDK) 1 or solar UV-induced signaling. The active AKT (p-AKT) mediates the signal transduction to downstream molecules such as the mTOR complex (mTORC)/p70S6 kinase (S6K), caspases, BCL2-associated agonist of cell death (BAD), mouse double minute 2 homolog (MDM2, E3 ubiquitin-protein ligase)/p53, forkhead box protein O1 (FOXO1)/B-cell lymphoma 2 (Bcl-2)*/*Bcl-2-like protein 11 (Bim)/Bcl-2-associated X protein (Bax), WEE1(G2 Checkpoint Kinase), p27 and glycogen synthase kinase 3 beta (GSK3*β*/*β*-catenin/glutamine synthetase (GS)/cyclin D1 for protein synthesis, cell survival, cell cycle, metabolism, migration and invasion (Figure 1). The other hands, PTEN/PI3K/AKT-mediated pathways promote cancer cell migration and invasion by the upregulation of matrix metalloproteinase (MMP)-2, MMP-9 and vascular endothelial growth factor (VEGF) [24,25]. Therefore, it is desirable to regulate the AKT kinase expression and activation and related signaling pathways for skin cancer therapy.

## 3. AKT and Related Signaling Pathway Inhibitors for Skin Cancer Regulation

Numerous phytomedicines derived from natural plant or fruit extracts exhibit anticancer activities against cell proliferation, survival, migration, metastasis and angiogenesis in vitro and in vivo. It has been previously reported that acacetin, isoangustone A, sulforaphane and tryptanthrin inhibited melanoma cell proliferation and tumor growth, and induced cell cycle arrest and apoptosis by directly or indirectly targeting PI3K/AKT/mTOR signaling pathways [26,27,28,29]. Here we describe the mechanisms of the latest natural inhibitors targeting AKT kinase and related signaling pathways in skin cancer preclinical studies in vitro and in vivo (Tables 1 and 2, Figure 3).

### 3.1. Isorhamnetin

The flavonoid 3′-methoxy-3,4′,5,7-tetrahydroxyflavone is derived from Persicaria thunbergii H. and Elaeagnus rhamnoides (L.) with reported anticancer effects on liver, colorectal, breast and lung cancers [30]. In melanoma, B16F10 cells, isorhamnetin treatment at 10 to 100 μM concentrations in different time points inhibited cell viability (72 h) and migration (12 and 24 h), and induced apoptosis (24 h) via downregulation of pAKT expression [31]. Mechanically, the 6-phosphofructo-2-kinase/fructose-2,6-bisphosphatase 4 (PFKFB4) metabolic enzyme was inhibited by isorhamnetin, which decreased pAKT expression and was verified by silencing the PFKFB4 expression. Additionally, treatment of C57BL/6 mice bearing B16F10 tumors with 20 mg/kg isorhamnetin retarded the growth and inhibited the expression of cell proliferation marker Ki-67.

### 3.2. Curcumol

C15H24O2-(3s-(3a,3aa,5a,6a,8ab))-octahydro-3-methyl-8-methylene-5-(1-methylethyl)-6h-3a,6-epoxyazulen-6-ol, is a polyphenol compound derived from Curcuma Wenyujin (ethanol fraction), with pharmacological anticancer activities in liver, lung and gastric cancer [32]. Curcumol inhibited B16 cell viability, colony formation and tumor growth, in vitro (at 50–200 μM) and in vivo (20 mg/kg, intraperitoneal (i.p.)), as well as migration and invasion at 50 and 100 μM and lung metastasis [33]. Regulation of miR-152-3p by curcumol treatment suppressed mesenchymal epithelial transition factor (c-MET) expression following the downstream signal pAKT expression.

### 3.3. Polyphyllin I

Diosgenyl alpha-L-rhamnopyranosyl-(1-2)-(beta-L-ara-binofuranosyl-(1-4)-beta-D-glucopyranoside), is a major component of Paris polyphylla and inhibits the growth of gastric and ovarian cancers, and osteosarcoma [34,35]. Treatment with polyphyllin I at 1.5, 3.0 and 6.0 mg/L concentrations suppressed A375 melanoma cell progression by reducing cell proliferation, migration and invasion, and the induction of G1 cell cycle accumulation, apoptosis and autophagy at 48 h [36]. Polyphyllin I-induced apoptosis and autophagy was mediated via the downregulation of pPI3K, pAKT and pmTOR expression. It was validated by A375 cell xenograft mouse experiments demonstrating that polyphyllin I (5 mg/kg, i.p.) inhibited tumor growth in both volume and weight. The levels of proliferation marker, Ki-67 and apoptosis feature, TUNEL-positive cells, were inhibited and increased in polyphyllin I-treated tissues compared with vehicle-treated controls, respectively.

### 3.4. Herbacetin

The flavonoid 3,5,7,8-tetrahydroxy-2-(4-hydroxyphenyl) chromen-4-one is a compound derived from flaxseed and ramose scouring rush, which suppresses breast and colon cancers and hepatocellular carcinoma [37]. Herbacetin showed anticarcinogenic and anticancer activities in A431 cutaneous squamous cell carcinoma and SK-MEL-5 melanoma in vitro at 10 and 20 μM concentrations, and in a mouse model of two-stages skin carcinogenesis induced by 7,12-dimethylbenz[α]-anthracene (DMBA)/12-O-tetradecanoylphorbol-13-acetate (TPA) and solar UV (48 kJ/m^2^ UVA, 2.9 kJ/m^2^ UVB) as well as in a SK-MEL-5 cell xenograft mouse model in vivo following topical treatment (100 and 500 nmol) and i.p. injection (0.2 and 1 mg/kg) [38]. Based on the AKT kinase assay, ex vivo pull-down assay and in silico docking analysis, herbacetin directly binds to AKT and inhibits its activity as well as the related signaling pathways such as GSK3β and RSK2.

### 3.5. Luteolin

The flavonoid 3′,4′,5,7-tetrahydroxyflavone is a compound derived from Reseda luteola, celery, broccoli and onions, and inhibits lung, breast and colon cancers [39,40]. Luteolin suppressed UVB (0.05 J/cm^2^)-induced COX-2 expression, AP-1 and NF-κB activation by the downregulation of pERK, pp38, pJNK and pAKT in JB6 P+ cells via directly targeting PKCε and c-Src kinase activity at 10 and 20 μM, and attenuated UVB (0.18 J/cm^2^)-induced skin tumorigenesis in SKH-1 hairless mice topically treated with 10 and 40 nmol [41]. Yao X et al. reported that luteolin inhibited A375 melanoma cell proliferation and induced apoptosis as well as migration and invasion at 10, 15 and 20 μM by decreasing MMP-2 and MMP-9 and increasing TIMP-1 and TIMP-2 expression through reducing the pPI3K and pAKT1 levels [42]. In the A375 cell xenograft mouse model, luteolin (100 mg/kg, i.p.) retarded the tumor growth and weight, and inhibited the expression of MMP-2 and MMP-9, and pPI3K and pAKT1.

### 3.6. Sinomenine

The alkaloid component 7,8-didehydro-4-hydroxy-3,7-dimethoxy-17-methylmorphinane-6-one is derived from Sinomenium acutum which is used for rheumatoid arthritis [43]. The anticancer activities of sinomenine have been reported in ovarian, gastric and lung cancers [31,44]. Sinomenine significantly inhibited caspase-3 activity and the viability of B16-F10 mouse melanoma cells and induced apoptosis and related proteins expression (Bax/Bcl-2 ratio) at 25, 50 and 100 μM in a CCK-8 assay and Annexin V stained cell counting as well as in Western blot analysis [45]. In detail, sinomenine increased autophagy by the induction of beclin1 expression and LC3II/LC3I level ratio and reduction in p-p62/SQSTM1 expression by the inhibition of pAKT and pmTOR expression, and thereby suppressed the cell growth and increased apoptosis/autophagy. The results were confirmed by treatment of cells with chloroquine, an autophagy inhibitor and a LY294002, PI3K/AKT inhibitor. Moreover, sinomenine (100 mg/kg, s.c.) retarded the melanoma tumor growth in vivo.

### 3.7. Syringic Acid

The active compound 4-hydroxy-3,5-dimethoxybenzoic acid is derived from Euterpe oleracea and Rhus javanica extract, and inhibits colorectal cancer and oral squamous cell carcinoma [46,47]. Syringic acid (0.2 and 1 mM) prevented UVB (0.2 J/cm^2^)-induced skin papilloma and tumors in SKH-1 hairless mice [48]. Mechanically syringic acid (40 μM) inhibited UVB (0.04 J/cm^2^)-induced reactive oxygen species (ROS) production by regulation of NADPH oxidase (NoX) activity and enhanced the interaction between phosphatase (PTP)-κ and epidermal growth factor receptor (EGFR), which was then decreased following signaling cascade pBraf and pAKT expression in human skin epithelial HaCaT cells.

### 3.8. Ginkgo Biloba Exocarp Extract

Derived from Ginkgo biloba L. (Ginkgo nuts) via the water solution–alcohol precipitation method, suppressed Lewis lung cancer cell growth or angiogenesis by regulation of mitogen-activated protein kinase (MAPK) signaling pathways and Wnt/β-catenin-vascular endothelial growth factor (VEGF) signaling pathways [49,50]. Ginkgo biloba exocarp extract inhibited B16-F10 cell proliferation, migration and adhesion to HUVEC cells in vitro at 10, 20 and 40 μg/mL by decreasing the expression of pPI3K and pAKT, and reduced B16F10 cells grafted tumor growth and lung metastasis in C57BL/6J mice, in vivo at 50, 100 and 200 mg/kg [51].

### 3.9. α-Mangostin

An active component of Garcinia mangostana Linn, 1,3,6-Trihydroxy-7-methoxy-2,8-bis(3-methylbut-2-en-1-yl)-9H-xanthen-9-one inhibits cervical cancer and hepatocellular carcinoma by the suppression of cancer stemness and signal transducer and activator of transcription (STAT3) signaling via Src homology region 2 domain-containing phosphatase (SHP)-1 stabilization [52,53]. α-Mangostin (5 and 10 mg/kg, i.p.) reduced the tumor incidence and the average numbers in a DMBA/TPA-induced skin carcinogenesis mouse model compared to a control group treated with olive oil [54]. The expression of proinflmmatory cytokines, IL-4, IL-18 and IL-1β was decreased and that of IL-10 was increased in both skin tumor and blood samples, and the level of apoptosis markers, cleaved PARP, cleaved caspase-3, Bad and Bax were induced and Bcl-xL and Bcl-2 were reduced by α-mangostin treatment. Moreover, pPI3K, pAKT and pmTOR expression were decreased by α-mangostin administration in mouse tissues of DMBA/TPA-induced skin carcinogenesis.

### 3.10. Silibinin (Silybin)

The bioactive compound (2R,3R)-3,5,7-trihydroxy-2-[(2R,3R)-3-(4-hydroxy-3-methoxyphenyl)-2-(hydroxymethyl)-2,3-dihydro-1,4-benzodioxin-6-yl]-2,3-dihydrochromen-4-one is derived from Silybum marianum (L.) and Gaertn. (Asteraceae) and inhibits melanoma and nasopharyngeal carcinoma by the downregulation of MEK/MAPK signaling pathways and PD-L1 expression [14,55]. Silibinin (25, 50 and 100 μM) and its 2,3-dehydro-derivative (DHS, 20 and 30 μM) suppressed ASZ001 (ASZ) and BSZ basal cell carcinoma cells proliferation and colony formation, and induced apoptosis [56]. Silibinin (50 and 100 μM) and DHS (50 and 100 μM) also inhibited signaling pathways including pEGFR, pERK1/2, pAKT and pSTAT3 expression in ASZ cells at 72 h, thereby abrogating NF-κB and AP-1 activities. Based on the results of ASZ cells in a mouse allograft model, the oral administration of silibinin (200 mg/kg) and DHS (200 mg/kg) significantly reduced the tumor weight. Dheeraj A et al., also reported that silibinin inhibited hedgehog inhibitor SANT-1 or GDC-0449- resistance ASZ001 basal cell carcinoma cell growth and colony formation and induced apoptosis by the regulation of pEGFR, pAKT and pERK expression [57].

### 3.11. Curcumin

The turmeric flavonoid compound 1,7-bis(4-hydroxy-3-methoxyphenyl)-1,6-heptadiene-3,5-dione is a from rhizome of Curcuma longa, regulating breast and bladder cancer development [58,59]. The proliferation and invasion of A375 and C8161 melanoma cells was inhibited by treatment with curcumin (25 or 15 μM), which induced G2/M phase cell cycle accumulation and autophagy by a reduction in pAKT, pmTOR and pP70S6K expression [60]. Additionally, curcumin (25 mg/kg, i.p.) suppressed the tumor growth of A375 cell-xenografted mice.

### 3.12. Other Bioactive Components

The viability of human uveal melanoma UM-1 cells was inhibited by treatment with pristimerin, a quinine methide triterpenoid compound derived from Celastraceae and Hippocrateaceae and the apoptosis induction was mediated by disrupting the mitochondrial membrane potential and increasing the ROS production. Furthermore, pristimerin reduced migration and invasion by the regulation of pAKT and pFoxO3a expression with confirming the knock-down of AKT in UM-1 cells [61]. In human A375 and mouse B16F10 melanoma cells, bioactive compounds such as gambogic acid [62], melittin [63], kaempferol [64], euplotin C [65], lycorine [66], oxyfadichalcone C [67], isoliquiritigenin [68], muniziqi granule/harmine [69], apigenin [70] and casticin [71] inhibited cell proliferation, colony formation, migration and invasion by downregulating the expression of pPI3K, pAKT, pmTOR and pGSK3*β* together with the expression of related biomarkers including p27, cyclin D1, LC3, 4EBP1, Bax, Bcl-2 and MMPs. A375.S2 cells, which are investigated in studies involving metastasis, chrysin [72] and berberine [73], significantly reduced cell mobility, migration and invasion by decreasing the level of MMPs, N-cadherin and uPA expression through the inhibition of PKC and pAKT. In A431 non-melanoma skin cancer cells, treatment with caffeic acid n-butyl ester induced G2/M phase of cell cycle arrest and apoptosis and inhibited migration by decreasing the expression of pPI3K, pAKT and pmTOR [74].

## 4. Perspectives

Multisteps of skin carcinogenesis are processed by initiation, promotion and progression [75]. UV is itself both initiator and promoter, and chemicals such as DMBA and TPA are initiators or initiator/promoters. Initiators trigger the DNA damage or ROS production. It can be removed by repair system in healthy cells, however, when the cells fail to recover from DNA damage or oxidative stress then the cells are transformed into neoplastic cells. This initiation and promotion steps indicated by inflammation markers including COX-2, NF-kB and AP-1, and PI3K/AKT/mTOR signaling pathways. Transformed cells continuously progress to cancer by proliferation and spread to other organs by migration and invasion. In these stages, PI3K/AKT/mTOR signaling pathways mediate to induce the survival and migration/invasion biomarkers including cyclins, Bcl-2 family and MMPs. Phytochemicals derived from plants can control each step of carcinogenesis, cancer proliferation and metastasis. In Table 1 and Table 2, most of natural compounds showed antiproliferation, antisurvival, antimigration and anti-invasion of skin cancers by the regulation of AKT-mediated signaling while syringic acid, herbacetin and α-mangostin inhibited DMBA/TPA or UV-induced skin carcinogenesis. Herbacetin directly targeted the ATP-binding pocket of the AKT catalytic domain and others are indirectly affected by AKT up- or downstream signaling pathways. Most effective phytochemicals contain the structure of flavonoid and polyphenols; however, it is difficult to dissolve in water and therefore limited to effect on target organs in vivo. Therefore, the formulation of natural compounds for improving the permeability could be modified such as ethosome. Binary ethosome of evodiamine and fisetin derived from Evodia rutaecarpa and onion enhanced the inhibitory activities of B16 melanoma cell proliferation and UVB-induced inflammation in mice [76,77]. Additionally, the combination therapy of clinical agents and natural compounds is expected to yield positive results. Application of natural compounds in skin cancer therapy requires the standardization of the plant-derived components and elucidation of their action mechanisms. On the other hand, we should consider other options with natural compounds mediated not only via canonical AKT-mediated signaling pathways but also new AKT-mediated signaling mechanisms such as miR-152-3p/c-MET/AKT and AKT/PFKFB4 pathways [31,33].

## Figures and Tables

**Figure 1 ijms-21-06869-f001:**
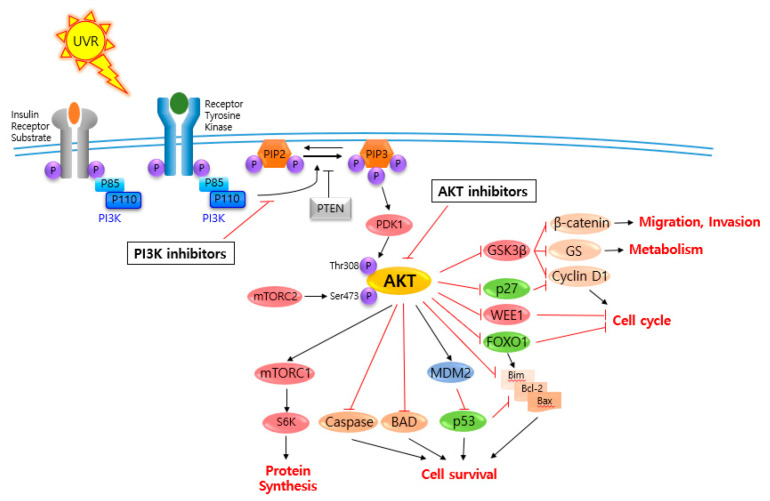
V-akt murine thymoma viral oncogene homolog (AKT) and related signaling pathways are associated with several features of skin cancer.

**Figure 2 ijms-21-06869-f002:**
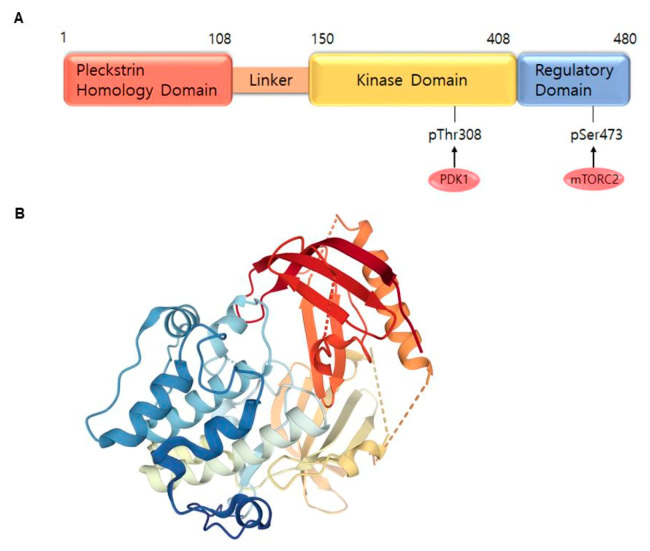
AKT1 structure. (**A**) Domains of the AKT1. AKT consists of an N-terminal pleckstrin homology (PH) domain (residues 5–108), linker, a catalytic kinase domain (residues 150–408) and a hydrophobic C-terminal tail (HM) (residues 409–480). (**B**) Crystal structure of AKT1 (PDB ID:3O96). Red indicates N-terminal domains consisting of PH; blue represents HM (https://www.rcsb.org/) (accessed on 18 August 2020).

**Figure 3 ijms-21-06869-f003:**
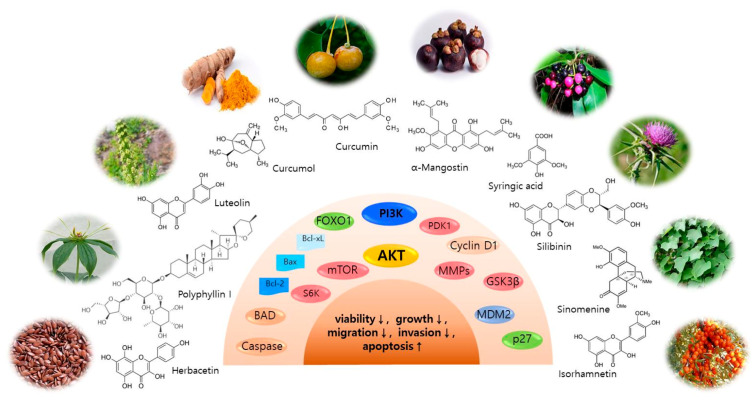
Natural compounds for effective treatment through various signaling factors in skin cancer therapy. Image Source: https://en.wikipedia.org/, https://www.naturalmedicinefacts.info/plant/sinomenium-acutum.html, https://www.wikiwand.com/it/Elaeagnus_rhamnoides, https://www.selleckchem.com, https://www.sigmaaldrich.com/. (accessed on 18 August 2020)

**Table 1 ijms-21-06869-t001:** List of natural compounds targeting PI3K/AKT/mTOR signaling pathway in various skin cancers (in vitro).

Compounds	Plants	Cancer Types	Cell Lines	Mechanisms	Ref.
**Isorhamnetin**	Persicaria thunbergii H., Elaeagnus rhamnoides (L.)	melanoma	B16F10	proliferation↓, migration↓ pAKT↓, PFKFB4↓	[31]
**Curcumol**	Curcuma wenyujin	melanoma	B16F10	viability↓, colony formation↓ migration↓, pAKT↓, c-MET↓, miR-152-3p↓	[33]
**Polyphyllin I**	Paris polyphylla	melanoma	A375	cells growth↓, migration↓, invasion↓, cell cycle progression↓, apoptosis↑, Bax↑, cleaved caspases-3↑, Bcl-2↓ autophagy↑, Beclin 1↑, LC3II ↑, p62 ↓ pPI3K↓, pAKT↓, pmTOR↓	[36]
**Herbacetin**	Flaxseed, ramose scouring rush herb	SCC, melanoma	JB6, A431, SK-MEL-5, SK-MEL-28	AKT1/2 activity↓, ODC activity↓, growth↓, neoplastic transformation↓, pGSK3β↓, ODC activity↓, AP1 activity↓, NF-κB activity↓, pERK1/2↓, p65↓	[38]
		melanoma	A375, Hs294T	tumor growth↓, angiogenesis↓, pEGFR↓, pAKT↓, pERK ↓ pGSK3β↓, ODC activity↓, AP1 activity↓, NF-κB activity↓	
**Luteolin**	Reseda luteola	melanoma	A375.S2	proliferation↓, migration↓, invasion↓, apoptosis↑, MMP-2↓, MMP-9↓, TIMP-1↑, TIMP-2↑, pAKT1↓, pPI3K↓	[41]
**Sinomenine**	Sinomenium acutum	melanoma	B16F10	cell viability↓, apoptosis↑, Bax↑, Bcl-2↓, caspase-3 activity↑, autophagy↑, Beclin-1↑, LC3II/LC3I ratio↑, pp62/SQSTM1↓, pAKT↓, pmTOR↓	[45]
**Syringic acid**	Euterpe oleracea, Rhus javanica	non-melanoma	HaCaT	UVB-induced COX-2↓, UVB-induced MMP-1↓, UVB-induced PGE2 generation↓, UVB-induced AP-1 activity↓, pERK1/2↓, pJNK1/2↓, pp38↓, pMEK1/2↓, p-MKK4/7↓, pMKK3/6↓, pB-Raf↓, pAKT↓, pSrc↓, EGFR↓, UVB-induced cyclooxygenase-2↓, matrix metalloproteinase-1↓, prostaglandin E2↓	[48]
**Ginkgo biloba Exocarp Extract**	Ginkgo biloba L.	melanoma	B16F10	proliferation↓, migration↓, heterogeneous adhesion↓, pPI3K↓, pAKT↓, NF-κB↓, MMP-9↓	[51]
**Silibinin**	Milk thistle plant (*Silybum marianum*)	BCC	ASZ001, Sant-1, GDC-0449 resistance ASZ001	growth↓, colony formation ↓, pEGFR↓, pAKT↓, cyclin D1↓, Gli-1↓, SMO↓, SUFU↓, apoptosis↑, caspase-3↑, Bcl-2↓	[56]
	*Silybum marianum* (L.) Gaertn., Asteraceae	BCC	ASZ, BSZ	cell growth↓, clonogenicity↓, apoptosis↑, pEGFR↓, pERK1/2↓, pAKT↓, pSTAT3↓	[57]
**Curcumin**	rhizome of Curcuma longa	melanoma	A375 and C8161	proliferation↓, invasion↓, G2/M phase cell-cycle arrest↑, autophagy↑, pAKT↓, pmTORC1↓, pp70S6K↓	[60]
**Pristimerin**	Celastraceae, Hippocrateacea	uveal melanoma	UM-1	apoptosis↑, viability↓, colony formation↓, mitochondrial membrane potential↓, ROS level↑, G0/G1 phase arrest↑ migration↓, invasion↓ pAKT↓, pFoxO3a ↓, Bim↑, p27^Kip1^↑, cleaved caspase-3↑, PARP↑, Bax↑, Cyclin D1↓, Bcl-2↓	[61]
**Gambogic acid**	resin of Garciania hanburyi	melanoma	A375, B16F10,	proliferation↓, migration↓, invasion↓, adhesion↓, EMT↓, angiogenesis processes↓ MMP-2 and MMP-9 activities↓ PI3K–AKT–mTOR signaling pathway↓	[62]
**Melittin** **/Bee Venom**	honey bees (*Apis mellifera*)	melanoma	B16F10, A375SM, SK-MEL-28	growth↓, colony-forming ability↓, migration↓, invasion↓, apoptosis↑, cleaved caspase-3 and -9↑, pPI3K↓, pAKT↓, mTOR↓, ERK↓, p38↓	[63]
**Kaempferol**	piper	melanoma	A375	proliferation↓, migration↓, colony formation↓, apoptosis↑, G2/M cell cycle arrest↑, pmTOR↓, pPI3K↓, pAKT↓	[64]
**Euplotin C**	*Euplotes crassus*	melanoma	A375, 501Mel, MeWo, HDFa	viability↓, apoptosis↑, migration↓, B-Raf↓, pERK 1/2↓, pAKT↓	[65]
**Lycorine**	*Lycoris radiate* spider lilies (Lycoris), daffodils (Narcissus) and snowdrops (Galanthus)	malignant melanoma	HEMa, A375	proliferation↓, cell migration↓, invasion↓, apoptosis↑, caspase-3↑, Bax↑, Bcl-2↓, pAKT↓, pmTOR↓, 4EBP1↓	[66]
**Oxyfadichalcone C**	Oxytropis falcate	melanoma	A375	proliferation↓, G1 phase arrest↑, apoptosis↑, migration↓, invasion↓, p27↑, cyclin D1↓, ppRb↓, pIntegrin β1↓, MMP-2/9↓, metastasis↓, pPDK1↓, pAKT↓, pGSK-3β↓, pmTOR↓, pp70s6k↓, pERK↓	[67]
**Isoliquiritigenin**	Glycyrrhizae Radix	melanoma	A375	proliferation↓, G2/M cell cycle arrest↑, mTOR↓, RICTOR↓, pAKT↓, pGSK-3β↓	[68]
**Muniziqi granule/harmine**	*Peganum harmala*, *Cichorium intybus*, *Dracocephalum moldavica*, *Ocimum basilicum*, *Althaea rosea*, and *Nigella glandulifera*	melanoma	B16F10	proliferation↓, autophagy, autophagosome formation↑, LC3-II↑, P62↓, apoptosis↑, G1 cell cycle arrest↑, pAKT↓, pmTOR↓, pERK1/2↓	[69]
**Apigenin**	Various fruits and vegetables		A375, C8161	proliferation↓, migration↓, invasion↓, apoptosis↑, G2/M cell cycle arrest↑, cleaved caspase-3↑, cleaved PARP↑, pERK1/2↓, pAKT↓, pmTOR↓	[70]
**Casticin**	Fructus viticis	melanoma	B16F10	migration↓, invasion↓, MMP-9↓, MMP-2↓, MMP-1↓, FAK↓, 14-3-3↓, GRB2↓, AKT↓, NF-κB↓, p65↓, SOS-1↓, p-EGFR↓, p-JNK 1/2↓, uPA↓, Rho A↓	[71]
**Chrysin**	passionflower, silver linden, honey, propolis	melanoma	A375.S2	mobility↓, migration↓, invasion↓, MMP-2 activity↓, GRB2↓, SOS-1↓, PKC↓, pAKT (Thr308)↓, NF-κBp65↓, NF-κBp50↓ uPA↓, N-cadherin↓, MMP-1↓, MMP-2↓, VEGF↓, E-cadherin↑, NF-κBp65↓	[72]
**Berberine**	the roots and bark of *Berberis* genus	melanoma	A375.S2	morphological changes↑, viability↓, mobility↓, migration↓, invasion↓, MMP-9 activity↓, MMP-1↓, MMP-13↓, E-cadherin↑, N-cadherin↓, RhoA↓, ROCK1↓, SOS-1↓, GRB2↓, Ras↓, pERK1/2↓, pc-Jun↓, p-FAK↓, pAKT↓, NF-κB↓, uPA↓, PKC↓, PI3K↓	[73]
**caffeic acid n** **-butyl ester**		skin carcinoma	A431	Apoptosis↑, Bax↑, Bcl-2↓, ROS↑, MMP↓, G2 phase arrest↑, migration↓, pmTOR↓, pPI3K, pAKT↓	[74]

↓; decrease ↑; increase.

**Table 2 ijms-21-06869-t002:** List of natural compounds targeting PI3K/Akt/mTOR signaling pathway in various skin cancers (in vivo).

Compounds	Plants	Cancer Types	Model	Treatment	Mechanisms	Ref.
**Isorhamnetin**	Hippophae rhamnoides L.	melanoma	C57BL/6 mice injected with B16F10 cells, 1 × 10^5^	20 mg/kg per day; for 7 days	Proliferation↓, Ki67↓	[31]
**Curcumol**	Curcuma wenyujin	melanoma	C57BL/6 mice injected (s.c. into the right lower paw and i.v. into the tail vein) with B16 cells, 2 × 10^6^	20 mg/kg, i.p.; 3 times per week; for 30 days	proliferation↓, growth↓ invasion↓, metastasis↓	[33]
**Polyphyllin I**	Paris polyphylla	melanoma	male BALB/c -nude mice with A375 cells, 2 × 10^6^	Polyphyllin I 5 mg/kg; i.p.; once a day for 35 days	tumor weight↓, tumor size↓ apoptosis↑, TUNEL positive cells↑, Ki67↓	[36]
**Herbacetin**	Flaxseed, ramose scouring rush herb	SCC	-DMBA/TPA model; Hairless SKH:HR-1-hrBr (SKH-1) (8–9 weeks old), initiation with DMBA (200 nmol), and promotion with 17 nmol of TPA in acetone, topically applied twice weekly for 20 weeks -solar–UV induced-skin carcinogenesis model; exposed to solar–UV (48 kJ/UVA/2.9 kJ/UVB) three times weekly for 12 weeks -xenograft model; Athymic mice (Cr:NIH(S), NIH Swiss nude, 6–9-wk-old) with SK-MEL-5 cells, 3 × 10^6^	-DMBA/TPA model; 100 or 500 nmol of herbacetin applied to dorsal mouse skin at 30 min before TPA treatment. -solar–UV induced-skin tumor mouse model; after 20 weeks later, herbacetin 100 or 500 nmol for an additional 7 weeks -xenograft model; herbacetin 0.2 and 1 mg/kg; i.p. injected three times per week for 15 days	skin papillomas↓, tumor volume↓, Ki67↓, pAKT↓, pGSK3β↓, pRSK↓, ODC↓	[38]
**Luteolin**	Reseda luteola	Melanoma	Female BALB/c -nude mice with A375 cells, 1 × 10^7^	100 mg/kg/day, i.p. for 22 days	tumor growth↓ PI3K/AKT↓, MMP-2↓, MMP-9↓	[41]
**Sinomenine**	Sinomenium acutum	Melanoma	xenograft model; BALB/c nude mice (6-week-old) by subcutaneously injection with B16-F10 cells	100 mg/kg/day; s.c., daily for 35 days.	tumor weight↓, tumor volume↓, Ki67↓, PCNA↓	[45]
**Syringic acid**	Euterpe oleracea, Rhus javanica	non-melanoma	SKH-1 hairless mouse, UVB (0.2 J/cm^2^) exposure (three times per week for 22 weeks)	0.2 or 1 mM per mouse in 200 μL acetone on the dorsal surface 1 h before UVB irradiation	UVB-induced skin tumor↓, COX-2↓, MMP-13↓,	[48]
**Ginkgo biloba Exocarp Extract**			C57BL/6J female mice (6-week-old) by subcutaneously injection with B16-F10, 2.0 × 10^6^ cells	50, 100, 200 mg/kg by intragestic gavage, once a day for 17 days	tumor growth↓, lung metastasis↓, MMP-9↓	[51]
**α-Mangostin**	pericarp of mangosteen	Skin cancer	DMBA (60 μg)/TPA (4 μg) induced skin carcinogenesis model in ICR female mice, once a week for 20 weeks	5 and 20 mg/kg, (dissolved in 0.2 mL olive oil) once a day, starting from the day after TPA was topically applied, i.p. for 20 weeks	Skin papilloma↓, growth↓, LC3↑, LC3-II↑, Beclin1↑, LC3-I↓, p62↓, Bax↑, cleaved caspase-3↑, cleaved PARP↑, Bad↑, Bcl-2↓, Bcl-xl↓, apoptosis↑, p-PI3K↓, p-AKT↓, p-mTOR↓	[54]
**Silibinin and its 2,3-dehydro-derivative**	*Silybum marianum* (L.) Gaertn., Asteraceae	BCC	ectopic allograft model; five weeks old nude mice (Foxn1^nu/nu^) by subcutaneously injection with 1 × 10^6^ ASZ cells	silibinin (200 mg/kg in 0.5% CMC) or DHS (200 mg/kg); oral administration, 6 days per week for a total of 7 weeks	tumor growth↓, PCNA↓, cyclin D1↓, proliferation↓, NF-κB↓, AP-1↓, c-Fos↓	[56]
**Curcumin**	rhizome of Curcuma longa	Melanoma	BALB/c nude female mice (6-week-old) by subcutaneously injection with A375 cells (1 × 10^7^/mL)	25 mg/kg by i.p. injections, every day for 3 weeks	growth↓	[60]

↓; decrease ↑; increase.

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
