# Peer review of "Alternative Options for Skin Cancer Therapy via Regulation of AKT and Related Signaling Pathways"

_ijms, 2020, doi:10.3390/ijms21186869_

Round 1

Reviewer 1 Report

The authors provided review for the use of phytomedicines to inhibit PI3K/Akt pathway in skin cancer. This is an interesting review focused on commonly overlooked topic of phytomedicines in cancer treatment. The manuscript requires extensive editing of English language and improvement in writing style (sections from 1-102 and from 235-247). The instances are too numeral to point out and rewrite/reorganization of the mentioned above manuscript sections is required. The authors do not need to change the ideas summarized in the sections but need to express them in a correct English language style.

Author Response

The authors provided review for the use of phytomedicines to inhibit PI3K/Akt pathway in skin cancer. This is an interesting review focused on commonly overlooked topic of phytomedicines in cancer treatment. The manuscript requires extensive editing of English language and improvement in writing style (sections from 1-102 and from 235-247). The instances are too numeral to point out and rewrite/reorganization of the mentioned above manuscript sections is required. The authors do not need to change the ideas summarized in the sections but need to express them in a correct English language style.

Response : Thank you very much for your valuable suggestions. We have extensively edited the manuscript and corrected the text from Harrisco which is a company for Scientific English Research Paper Editing Service (certificate provided).

Reviewer 2 Report

The Review article of Dr Sun-Young Hwang et al aims to summarize the current state of understanding of the efficiency of plant extracts in the treatment of skin cancer. After a brief introduction concerning the different type of skin cancers and their etiology, the manuscript provides an overview of the PI3K/ Akt/ mTor signaling pathways. Then, the Authors describe the antitumor activity of a series of plant extracts, their molecular characteristics, their biological targets and their impact on skin cancer cell lines and/ or on experimental models of skin cancer in mice.

Considering the increasing incidence of skin cancers, their prognosis, their resistance to conventional and targeted therapy, the identification and characterization of the antitumor activity of novel bioactive compounds is of uppermost importance.

This report is of interest, nevertheless, I would like to raise the following points:

In the Introduction section, the Authors should describe in more detail the therapies currently in use for the treatment of skin cancer and their limitations (rather than in the perspectives). In this connection, the Authors should introduce the genetic defects involved in skin carcinogenesis before the treatment in use.

In my mind, the perspectives should delineate which steps are required, which problems have to be overcome for the transfer of phytomedicine from the bench to the bedside.

The results presented in the manuscript deal with the use of plant extracts in the treatment of skin cancer. Antitumor and chemopreventive activities clearly involve distinct biological mechanisms (e.g. inhibition of cancer cell proliferation, migration, induction of apoptosis vs chemoprotection of healthy cells from DNA damages,…). It might be interesting to mention these different aspects, (e.g. in the Introduction section or in the Perspectives), and briefly exemplify the potential chemopreventive activity of some compounds described in the manuscript or provide the references of some Reviews on this topic. Especially, skin chemoprotection using appropriate posology of plant extracts might be more easy to operate.

Line 60, the Authors indicate that PH domains are involved in protein-protein interaction. PH domains also interact with lipids e.g. phosphatidylinositol (3,4,5)-trisphosphate and trigger protein recruitment to the plasma membrane close to their effector systems (see below).

Lines 68-72: “In skin cancer, active form phospho-AKT (pAKT) was overexpressed in 22 (54%) 68 of 41 benign nevi and 112 (71.3%) of 157 primary melanoma tumors compared to normal adjacent 69 tissue, and caused to reciprocal five-year survival rate in metastatic melanoma [21,22]. The expression of activated AKT1 induces highly metastatic melanomas to lung (67%) and brain metastases (17%), 71 respectively together with downstream target protein mTOR [23].” The first sentence clearly concerns human melanoma, but the 2nd sentence reports results obtained with experimental mouse models. This should be clearly mentioned, furthermore the experimental mouse model should be more detailed.

The activation of Akt (line 73) should be more precisely explained. PI3K produces phosphatidylinositol (3,4,5)-trisphosphate that recruits Akt to the plasma membrane through its PH domain. The description of Akt effector systems (lines 74-84) and in particular the biological activity of the mTORC1/2 complexes should adopt a more explanatory approach.

In Figure1, the inhibitory effect of GSK3 on B-catenin (Wnt pathway) could be mentioned

The Authors could propose putative mechanisms of action of phytochemical, e.g. direct effect by targeting some effectors of the PI3K/ Akt/ mTor pathways (e.g. as mentioned in the text the interaction of herbacetin with Akt), or indirect mechanisms, such as modulation of gene/ miRNA expression; decreased ROS production. ROS are known among others to inactivate phosphatases (including PTEN that counteracts PI3K activity) through reversible oxidation.

For biological scientists that do not have a deep chemical background, the Authors could explain the overall classes of phytochemical compounds and putative targets.

The Author may mention that they describe in the text compounds that prove to exert antitumor activity on skin cancer either in experimental model in vivo or both in vivo and in vitro, and that a more complete list of phytochemical that proved in vitro efficiency is displayed in Table I.

It is not clear for me why the order of some compounds in the text and in the Table I changed.

Other phytochemicals have been shown to target the PI3K/ Akt/ mTor pathways, to exert biological effects on skin cancer cells, and should be mentioned: Acacetin, Capsaicin, Evodiamine, Fisetin, Isoangustone A, Sulforaphane, Tryptanthrin.

The manuscript requires an extensive editing of English language (please find below few examples):

Line 22,23 The sentence “Although BRAF inhibitors are treated for skin cancer (melanoma) targeted therapy, it needs to more options cause the resistance and weak efficacy.” should be rewritten.

Line 24: change “regulators” to “inhibitors”

The last sentence of the abstract should be rewritten to mention the focus of the Review on the interest of phytotherapy to target PI3K/AKT/mTor signaling pathway for the treatment skin cancer

Line 30 change “popular” to “common” or “frequent”

Line 31 “According to the cellular development,” should be change to “According to their cellular origin,”; “skin cancer mainly divided” to “skin cancers are mainly divided”

Line 33 “Among the all of skin cancers” to “Among all skin cancers”

Author Response

The Review article of Dr Sun-Young Hwang et al aims to summarize the current state of understanding of the efficiency of plant extracts in the treatment of skin cancer. After a brief introduction concerning the different type of skin cancers and their etiology, the manuscript provides an overview of the PI3K/ Akt/ mTor signaling pathways. Then, the Authors describe the antitumor activity of a series of plant extracts, their molecular characteristics, their biological targets and their impact on skin cancer cell lines and/ or on experimental models of skin cancer in mice.

Considering the increasing incidence of skin cancers, their prognosis, their resistance to conventional and targeted therapy, the identification and characterization of the antitumor activity of novel bioactive compounds is of uppermost importance.

This report is of interest, nevertheless, I would like to raise the following points:

In the Introduction section, the Authors should describe in more detail the therapies currently in use for the treatment of skin cancer and their limitations (rather than in the perspectives). In this connection, the Authors should introduce the genetic defects involved in skin carcinogenesis before the treatment in use.

Response 1 : Thank you very much for your valuable suggestion. We have moved the strategies of skin cancer treatment to introduction and added the genetic defects.

In my mind, the perspectives should delineate which steps are required, which problems have to be overcome for the transfer of phytomedicine from the bench to the bedside.

Response 2 : Thank you very much. We have explained that skin carcinogenesis processes and the chemopreventive and anticancer activities of phytomedicines, in vitro and in vivo, then suggested the options for overcoming the limitation of phytomedicines in perspectives.

The results presented in the manuscript deal with the use of plant extracts in the treatment of skin cancer. Antitumor and chemopreventive activities clearly involve distinct biological mechanisms (e.g. inhibition of cancer cell proliferation, migration, induction of apoptosis vs chemoprotection of healthy cells from DNA damages,…). It might be interesting to mention these different aspects, (e.g. in the Introduction section or in the Perspectives), and briefly exemplify the potential chemopreventive activity of some compounds described in the manuscript or provide the references of some Reviews on this topic. Especially, skin chemoprotection using appropriate posology of plant extracts might be more easy to operate.

Response 3 : Thank you for your kind suggestion. We have explained the skin carcinogenesis and cancer, and anticancer and chemopreventive activities of phytomedicines including action mechanisms in perspectives of manuscript.

Line 60, the Authors indicate that PH domains are involved in protein-protein interaction. PH domains also interact with lipids e.g. phosphatidylinositol (3,4,5)-trisphosphate and trigger protein recruitment to the plasma membrane close to their effector systems (see below).

Response 4 : Thank you. We have added it in the text (line 65-69).

Lines 68-72: “In skin cancer, active form phospho-AKT (pAKT) was overexpressed in 22 (54%) 68 of 41 benign nevi and 112 (71.3%) of 157 primary melanoma tumors compared to normal adjacent 69 tissue, and caused to reciprocal five-year survival rate in metastatic melanoma [21,22]. The expression of activated AKT1 induces highly metastatic melanomas to lung (67%) and brain metastases (17%), 71 respectively together with downstream target protein mTOR [23].” The first sentence clearly concerns human melanoma, but the 2nd sentence reports results obtained with experimental mouse models. This should be clearly mentioned, furthermore the experimental mouse model should be more detailed.

Response 5 : Thank you for your comments. We have put more clearly as to whether the experimental results were obtained from humans or mice as well as added the mouse model in detail (line 75-80).

The activation of Akt (line 73) should be more precisely explained. PI3K produces phosphatidylinositol (3,4,5)-trisphosphate that recruits Akt to the plasma membrane through its PH domain. The description of Akt effector systems (lines 74-84) and in particular the biological activity of the mTORC1/2 complexes should adopt a more explanatory approach.

Response 6 : Thank you very much. We have mentioned the AKT activation in detail.

In Figure1, the inhibitory effect of GSK3 on B-catenin (Wnt pathway) could be mentioned

Response 7 : Thank you for suggestion. We have added b-catenin/Wnt pathway for migration and invasion in the figure and text

The Authors could propose putative mechanisms of action of phytochemical, e.g. direct effect by targeting some effectors of the PI3K/ Akt/ mTor pathways (e.g. as mentioned in the text the interaction of herbacetin with Akt), or indirect mechanisms, such as modulation of gene/ miRNA expression; decreased ROS production. ROS are known among others to inactivate phosphatases (including PTEN that counteracts PI3K activity) through reversible oxidation.

Response 8 : We have mentioned the direct or indirect regulation of PI3K/ Akt/ mTOR pathways with phytochemicals in the manuscript.

For biological scientists that do not have a deep chemical background, the Authors could explain the overall classes of phytochemical compounds and putative targets.

Response 9 : Thank you for your kind suggestion. We have added the information of characters of compound moiety.

The Author may mention that they describe in the text compounds that prove to exert antitumor activity on skin cancer either in experimental model in vivo or both in vivo and in vitro, and that a more complete list of phytochemical that proved in vitro efficiency is displayed in Table I.

It is not clear for me why the order of some compounds in the text and in the Table I changed.

Response 10 : Sorry for confusing you. We have changed the order of compounds with matching the Table and text.

Other phytochemicals have been shown to target the PI3K/ Akt/ mTor pathways, to exert biological effects on skin cancer cells, and should be mentioned: Acacetin, Capsaicin, Evodiamine, Fisetin, Isoangustone A, Sulforaphane, Tryptanthrin.

Response 11 : Thank you, we have added the phytochemicals; evodiamine and fisetin with ethosome formulation in perspective and anti-skin cancer activities of acacetin, soangustone A, sulforaphane and tryptanthrin showed in introduction since we described recent five- year publications.

The manuscript requires an extensive editing of English language (please find below few examples):

Response 12 : Thank you for your comments. We have corrected and rewrite the manuscript and edited by Harrisco, a company for Scientific English Research Paper Editing Service (certificate provided).

Line 22,23 The sentence “Although BRAF inhibitors are treated for skin cancer (melanoma) targeted therapy, it needs to more options cause the resistance and weak efficacy.” should be rewritten.

Response 13 : We have changed the sentence, ‘Although BRAF inhibitors are used to treat melanoma further options are needed due to treatment resistance and poor efficacy.’

Line 24: change “regulators” to “inhibitors”

Response 14 : Thank you, we have changed ‘regulators’ to ‘inhibitors’.

The last sentence of the abstract should be rewritten to mention the focus of the Review on the interest of phytotherapy to target PI3K/AKT/mTor signaling pathway for the treatment skin cancer

Response 15 : Thank you, we have changed the sentence.

Line 30 change “popular” to “common” or “frequent”

Response 16 : Thank you, we have changed ‘popular’ to ‘frequent’.

Line 31 “According to the cellular development,” should be change to “According to their cellular origin,”; “skin cancer mainly divided” to “skin cancers are mainly divided”

Response 17 : According to your suggestion, we have changed it.

Line 33 “Among the all of skin cancers” to “Among all skin cancers”

Response 18 : Thank you very much. We have corrected grammars according to your suggestion.

Round 2

Reviewer 1 Report

Thank you for the revisions; I'm satisfied with the revisions.

Here are several minor points

Line 83. "It's also"

Line 84. "The active AKT (p-AKT) mediates"

Line 90. Contextually referencing Figure 1 is more appropriate here. Authors can decide whether to change or keep the reference. "synthesis, cell survival, cell cycle, metabolism, migration and invasion (Figure 1).

Line 256. "In Table 1 and 2". Table starts with capital letter

Reviewer 2 Report

The Authors have taken into consideration my comments and recommendations, and have change the manuscript accordingly.

I consider that this revised version is acceptable for publication.

Sincerely yours